# Prevalence, variety, and iron and zinc content of commercial infant and toddler foods sold in the United States that contain meat

Kameron J. Moding[1]*, Megan C. Lawless[2], Catherine A. Forestell[3], Katherine J. Barrett[2], Susan L. Johnson [ORCID][2]*

**1** Human Development and Family Science, Purdue University, West Lafayette, IN, United States of America, **2** Department of Pediatrics, Section of Nutrition, University of Colorado Anschutz Medical Campus, Aurora, CO, United States of America, **3** Psychological Sciences, William and Mary, Williamsburg, VA, United States of America

* Susan.Johnson@cuanschutz.edu (SLJ); kjmoding@purdue.edu (KJM)

**Data Availability Statement:** Data described in the manuscript and the associated codebook are available on Open Science Framework: https://doi.org/10.17605/OSF.IO/A2BFC.

## Abstract

Iron and zinc are important nutrients during infancy, particularly for infants exclusively fed human milk at the beginning of complementary feeding (CF) from 6–12 months. The *1st Foods Study* examined the ingredients and nutrient contents of commercially-available infant and toddler foods (ITFs) that were sold in the US and contained meat. Company websites ($n = 22$) were used to create a database of commercial ITFs ($n = 165$) available for purchase in the US and contained at least one meat (e.g., beef, chicken, pork). Single ingredient and ready-to-serve meals (for ages ≤ 9 months) and ready-to-serve meals (for ages 10+ months) were categorized as infant and toddler products, respectively. For each product, the ingredient list, intended age/stage, serving size (g), energy (kcal), protein (g), iron (mg), and zinc (mg) per serving were recorded from product labels. Nutrient amount/100 g was calculated for each product and medians and inter-quartile ranges were calculated and compared (1) by intended age/stage of the product and (2) according to meat type. In general, toddler products contained more iron than infant products. Within infant products ($n = 65$), more iron was found in products containing beef relative to products with other meats, which were similar in iron content. Within toddler products ($n = 38$), more iron was found in products containing seafood, followed by beef, turkey, and pork. Slightly less iron was found in products with chicken. Zinc content was infrequently reported ($n = 17$ total products). Because many of the products assessed contained low amounts of iron and zinc, meeting the current infant and toddler requirements for iron and zinc during the CF period may be challenging if commercial ITFs containing meat are the primary source of these nutrients.

## Introduction

The 2020–2025 Dietary Guidelines for Americans (DGA) includes recommendations for infants and toddlers under two years of age for the first time [1]. To ensure that nutrient and calorie needs are met to support the rapid growth and development that occurs during this period, it is recommended that infants and toddlers consume a variety of foods from all food

**Funding:** "Susan L. Johnson, Kameron J. Moding, Catherine Forestell, and Katherine J. Barrett report that financial support was provided for this study by the research program coordinated by the National Cattlemen's Beef Association, a contractor to the Beef Checkoff Program. Contract # 25B0508. The funders had no role in study design, data collection and analysis, decision to publish, or preparation of the manuscript.".

**Competing interests:** "Susan L. Johnson, Kameron J. Moding, Catherine Forestell, and Katherine J. Barrett report that financial support was provided for this study by the research program coordinated by the National Cattlemen's Beef Association, a contractor to the Beef Checkoff Program. Contract # 25B0508. This does not alter our adherence to PLOS ONE policies on sharing data and materials."

**Abbreviations:** CF, Complementary Feeding; DGA, Dietary Guidelines for Americans; DV, Daily Value; EAR, Estimated Average Requirement; FDA, Food and Drug Administration; FITS, Feeding Infants and Toddlers Study; ITF, Infant and Toddler Food; RACC, Reference Amount Customarily Consumed per eating occasion; US, United States; WIC, Special Supplemental Nutrition Program for Women, Infants, & Children.

groups, including vegetables, fruits, dairy, grains, and foods high in protein. The most recent data from the U.S. Feeding Infants and Toddlers Study (FITS) indicates that grains are widely consumed by infants and toddlers, followed by fruits and vegetables, but meat and other protein sources are consumed at lower levels [2]. Specifically, only 41% of 6- to 11.9-month-olds consume any meat or other protein sources (not including infant milks) on a given day, with only 30% consuming baby food meats or other meats. More toddlers consume these foods with 88% of 12- to 17.9-month-olds and 91% of 18- to 23.9-month-olds, consuming some protein sources (and 69% and 73%, respectively consuming meat) [2]. These low levels of meat consumption may be of concern because meat is a good source of bioavailable iron and zinc, two important nutrients for infants and toddlers [1].

It is important for infants and toddlers to consume iron-rich foods, such as meat, seafood, and beans/lentils, to support immune function and neurologic development, starting around 6 months of age [1]. The estimated average requirement (EAR) for iron is greater for infants (6.9 mg/day for 7- to 12-month-olds) than it is for toddlers (3.0 mg/day for 1- to 3-year-olds) [3]. By around 6 months of age, infants' prenatal iron stores are depleted and iron intake from exogenous dietary sources is often insufficient, particularly for the exclusively human milk fed infant [4]. Around the same age, infants should also begin consuming zinc-rich foods, such as meat, poultry, and to a lesser extent beans and legumes, to support immune function and growth [1]. Unlike iron, the EAR for zinc is consistent across infancy and toddlerhood at 2.5 mg/day [3]. Consumption of iron- and zinc-rich foods is critically important for infants fed human milk starting at 6 months because human milk alone does not provide a sufficient amount of iron or zinc; the content of both nutrients in human milk is low and cannot be improved by increasing intakes through the maternal diet [5]. As a result, an estimated three-quarters of infants fed human milk in the U.S. consume an inadequate amount of iron and an estimated half consume an inadequate amount of zinc to meet recommended intakes [1]. Intake of these nutrients is less of a concern for formula-fed infants due to their consumption of iron- and zinc-fortified formulas.

One potential strategy to improve intake of iron and zinc among infants and toddlers is to increase their consumption of iron- and zinc-rich foods such as meat, seafood, and poultry. These foods contain heme-iron which is more bioavailable than the non-heme iron found in plant-based foods [3]. In the U.S., the commercial infant and toddler food (ITF) market contains meat-based meals and snacks, but very few of these products are good or excellent sources of iron [6]. Further examination is needed to determine which of these products (e.g., products containing specific meat types, infant vs. toddler products) are more micronutrient dense. Information on the best sources of iron and zinc among commercially available meat-based meals and snacks could be helpful for caregivers concerned about the adequacy of their infant's or toddler's iron and zinc intake.

The present study was exploratory and had two aims. First, we aimed to identify the prevalence and variety of products containing meat generally, and by intended age or stage of the target audience (e.g., 12 months+, Stages 1–4). Second, we aimed to compare the nutrient contents, specifically energy (kcal), protein, iron, and zinc per serving and per 100 grams of meat-containing products intended for infants and toddlers.

## Methods

### Study overview

The present study addressed one set of primary aims from the *1st Foods Study*, which is a multiple methods study exploring caregiver decision-making related to introducing animal-source proteins to infants and toddlers. Information on the present study and analysis were pre-registered on Open Science Framework: https://doi.org/10.17605/OSF.IO/A2BFC.

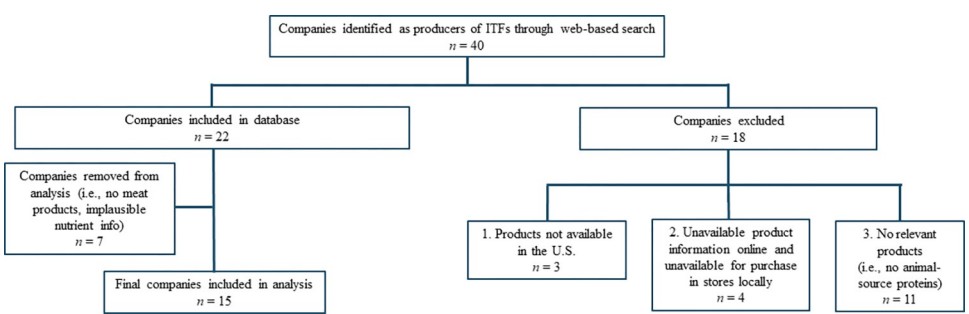

**Fig 1. Decision tree of processes for exclusion of infant and toddler food companies included in the present study.**

## Data collection

A web-based search of companies selling ITFs was conducted between July and November 2021. Companies were included if they had public websites containing ingredient lists and nutrition labels and sold products that contained animal-source proteins (e.g., meat, cheese) and were available for purchase online, in-stores, or through subscription services in the United States. For one company that had incomplete nutrient information listed online, an in-store shelf inventory was conducted. **Fig 1** depicts a decision tree for exclusion of companies. In total, 22 companies sold products containing animal-source proteins and were included in our database. One company was excluded from the present analysis because its products contained nutrient information that was deemed implausible (i.e., the energy content did not align with the macronutrient information on the Nutrition Facts panel). Given our present focus on iron and zinc, only companies selling products that contained meat (i.e., red meat, poultry, pork, seafood, or other meat) were selected and included for this analysis ($n = 15$).

All products containing the name of a specific meat (e.g., turkey, beef) or reference to meat (e.g., "Italian meatballs", "pot roast") were entered in the database. A wide variety of products were included, such as meat and gravy purées, purées with meat and vegetables, meat sticks or nuggets, and toddler meals containing meat (e.g., pasta with meatballs, meatloaf, or soups with meat).

## Product information

For each product, the following information was recorded: company, brand, product name, and ingredient list. The following information was also recorded from Nutrition Facts panels: serving size (g), and energy (kcal), protein (g), iron (mg), and zinc (mg) per serving. The amount of each nutrient per 100 g was also calculated for each product.

In addition, each product's intended age and/or stage was recorded from product labels. Intended age of the product is the youngest age, in months, that the product is intended to target (e.g., 9 months +). Although definitions of intended stage differ somewhat by company, most companies use definitions similar to the following: Stage 1: finely puréed, smooth "first" foods meant for infants about 4 months of age and older; Stage 2: strained foods for infants about 6 months of age and older; Stage 3: partially strained foods with small chunks for infants about 7–9 months and older; Stage 4: all foods meant for toddlers ages 12 months and older [7, 8].

Following prior work [6, 8, 9], products were categorized, when possible, based on relevant food categories outlined in the Food and Drug Administration's (FDA) guidance on the reference amount customarily consumed per eating occasion (RACC). In the present study, four RACC categories were applicable to products in the database: 1) Dinners, desserts, fruits,

vegetables, or soups, ready-to-serve, junior type (RACC = 110 g); 2) Dinners, desserts, fruits, vegetables, or soups, ready-to-serve, strained type (RACC = 110 g); 3) Dinners, stews, or soups for young children, ready-to-serve (RACC = 170 g) [10]; and 4) Plain meats and meat sticks, ready-to-serve (RACC = 55 g) [11]. Products meeting the criteria for categories 1 and 2 (RACC = 110 g) were categorized as *infant products* and products meeting the criteria for category 3 (RACC = 170 g) were categorized as *toddler products*. Some products (*n* = 17) were intended for infants (10 months +) but included a serving size (e.g., 171 g) and ingredients and/or textures indicative of a toddler product, so these products were categorized as toddler products. Products meeting the criteria for category 4 were analyzed separately as *meat snacks*.

A substantial number of products (*n* = 55) could not be categorized using the RACC categories above. Most of these products were marketed toward children, but their product descriptions mentioned that they could be offered to children as young as 12 months of age. The serving sizes of these products were highly variable (range 24 g– 245 g) and tended to be larger than typical toddler serving sizes (40% or *n* = 22 products had serving sizes larger than the toddler RACC size of 170 g). For these reasons, this group of products was analyzed separately from infant and toddler products and was labeled *young child products*.

Ingredient lists were used to categorize the ingredients in each product. Specifically, the presence (yes/no) of each of the following meats was recorded: beef, chicken, turkey, pork (including ham or bacon), seafood, and other meats. The presence (yes/no) was also noted for the following ingredients: dairy (including yogurt, milk, and cheese), eggs, grains, vegetables, and fruit. Vegetables were further classified into red/orange vegetables (e.g., carrots, sweet potatoes) or other vegetables using the DGA food pattern categories [1]; for our purposes "other vegetables" included all vegetables not categorized as red/orange, including dark green vegetables, beans/peas/lentils, starchy vegetables, and other vegetables (e.g., beets, green beans) [1]. Results are presented separately for red/orange vs. other vegetables because red/orange vegetables are the most commonly included vegetable type in ITFs [8].

To ensure accuracy of data entry from manufacturers' websites and product labels and ingredient coding, all product information and codes were double-entered for each product and compared. Discrepancies in initial data entry were resolved by checking websites and product labels and disagreements in ingredient coding were resolved through consensus.

## Statistical analysis

Frequencies of appearance for specific meats (e.g., beef, chicken) were calculated across all products by intended age and/or stage. Prevalence of products with meat as the sole ingredient and the presence of other ingredient types (i.e., eggs, dairy, grains, red/orange vegetables, other vegetables, and fruit) were determined for infant, toddler, and young child products. Then, the nutrient contents were examined. Descriptive statistics were calculated and distributions were analyzed by tests of normality for all nutrient variables (i.e., energy, protein, iron, and zinc per serving and per 100 grams). Given that most variables were non-normally distributed, medians and inter-quartile ranges were calculated and presented (1) for infant, toddler, and young child products and meat snacks, and (2) according to meat type. Inter-quartile ranges could not be calculated when a product category contained 3 products or fewer and, thus, are not reported.

Following previous research [6] the number of products considered to be a "good" source of iron was calculated. According to the FDA, a product is considered a "good" source of iron if it contains 10–19% of the daily value (DV) for iron [12]. Using this definition and the DVs for infants and toddlers [13], the number of infant and toddler products classified as a "good" source of iron were calculated. Since DVs are not as commonly used with infants and toddlers,

the same calculation (10–19%) was performed using the EAR for iron for infants and toddlers [3].

## Results

### Prevalence and variety of products containing meat (Aim 1)

In total, 165 products intended for infants/toddlers contained meat. **Table 1** displays the frequency of products by meat type across intended age and/or stage of the product. Most of the products containing meat were intended for infants ($n = 65$; 39%), followed by young children ($n = 55$; 33%), and toddlers ($n = 38$; 23%). A small number of products ($n = 7$; 4%) were categorized in RACC category 4: plain meats and meat sticks, ready-to-serve; of these, 2 products were intended for toddlers (12 months+) and 5 were intended for young children (24 months +). Across all meat-containing products, almost half contained chicken ($n = 80$; 49%), followed by beef ($n = 35$; 21%) and turkey ($n = 34$; 21%), and fewer products contained pork ($n = 11$; 7%) or seafood ($n = 7$; 4%). The distribution of products within each age group by meat type tended to mirror the overall percentages across all products.

Table 2 displays the frequency of products containing ingredients besides meat. Few infant products ($n = 9$; 14% of infant products) and no toddler products, young child products, or meat snacks contained meat as the sole ingredient in the product. Instead, most meat-containing products included a variety of other ingredients. Infant products commonly contained red/orange and other vegetables, followed by grains and fruit. Most toddler products contained other vegetables, grains, and/or red/orange vegetables, and a little less than half also contained egg and/or dairy. Almost all young child products contained other vegetables and/

**Table 1. Meat types represented in infant and toddler products by intended age or stage of the product[1].**

|  | Total n (%) | Beef[2] n (%) | Chicken n (%) | Pork/Ham n (%) | Seafood n (%) | Turkey n (%) |
|---|---|---|---|---|---|---|
| Infant | 65 (39) | 16 (25) | 28 (43) | 2 (3) | 4 (6) | 15 (23) |
| Stage 1 (4 months+) | 5 (8) | 1 (20) | 2 (40) | 0 (0) | 0 (0) | 2 (40) |
| Stage 2 (6 months+) | 25 (38) | 5 (20) | 13 (52) | 2 (8) | 0 (0) | 5 (20) |
| Stage 3: (9 months+) | 18 (28) | 4 (22) | 7 (39) | 0 (0) | 2 (11) | 5 (28) |
| No Stage Identified | 17 (26) | 6 (35) | 6 (35) | 0 (0) | 2 (12) | 3 (18) |
| Toddler | 38 (23) | 9 (24) | 18 (47) | 2 (5) | 2 (5) | 7 (18) |
| 10 months+ [3] | 5 (13) | 1 (20) | 3 (60) | 1 (20) | 0 (0) | 0 (0) |
| Stage 4 (12 months+) | 10 (26) | 2 (20) | 6 (60) | 0 (0) | 0 (0) | 2 (20) |
| No Age/Stage Identified | 23 (61) | 6 (26) | 9 (39) | 1 (4) | 2 (9) | 5 (22) |
| Young Child[4] | 55 (33) | 9 (16) | 29 (53) | 7 (13) | 1 (2) | 11 (20) |
| 12 months+ | 42 (76) | 9 (21) | 22 (52) | 7 (17) | 1 (2) | 5 (12) |
| No Age/Stage Identified | 13 (24) | 0 (0) | 7 (54) | 0 (0%) | 0 (0) | 6 (46) |
| Meat Snacks | 7 (4) | 1 (14) | 5 (71) | 0 (0) | 0 (0) | 1 (14) |
| 12 months+ | 2 (29) | 0 (0) | 1 (50) | 0 (0) | 0 (0) | 1 (50) |
| 24 months+ | 5 (71) | 1 (20) | 4 (80) | 0 (0) | 0 (0) | 0 (0) |
| Total | 165 | 35 (21) | 80 (49) | 11 (7) | 7 (4) | 34 (21) |

[1]Percentages were calculated for each meat type out of the total number of products within each intended age/stage category (e.g., Stages 1–4, 10 months+)

[2]One product containing bison was included in the beef category.

[3]These products were intended for infants but included a serving size and ingredients and/or textures indicative of a toddler product, so they were analyzed with the toddler products.

[4]Products containing more than one meat ($n = 2$) are represented in more than one column.

**Table 2. Other ingredients included in infant and toddler products containing meat[1].**

| | Total n | Single ingredient n (%) | Red/orange vegetable n (%) | Other vegetable[2] n (%) | Fruit n (%) | Grains n (%) | Dairy n (%) | Egg n (%) |
|---|---|---|---|---|---|---|---|---|
| Infant (RACC 110g) | 65 | 9 (14) | 46 (71) | 44 (68) | 24 (37) | 36 (55) | 2 (3) | 0 (0) |
| Toddler (RACC 170g) | 38 | 0 (0) | 31 (82) | 34 (90) | 7 (18) | 31 (82) | 16 (42) | 16 (42) |
| Young Child (No RACC) | 55 | 0 (0) | 39 (71) | 53 (96) | 14 (26) | 52 (95) | 39 (71) | 22 (40) |
| Meat Snacks (RACC 55 g) | 7 | 0 (0) | 0 (0) | 7 (100) | 0 (0) | 5 (71) | 0 (0) | 1 (14) |
| Total | 165 | 9 (6) | 116 (70) | 138 (84) | 45 (27) | 124 (75) | 57 (35) | 39 (24) |

*Note*. RACC = reference amount customarily consumed per eating occasion.

[1]Percentages were calculated within RACC categories (rows) for infant, toddler, young child, and meat snack products. The percentage of products containing each ingredient was calculated out of the total number of products within each RACC category. Products containing more than one ingredient besides meat are included in more than one column.

[2] "Other vegetable" includes all vegetables not categorized as red/orange, including dark green vegetables, beans/peas/lentils, starchy vegetables, and other vegetables (e.g., beets, green beans).

or grains and many contained dairy or red/orange vegetables. Finally, all meat snacks (*n* = 7) contained other vegetables (mostly in the form of onion and/or garlic powder) and commonly contained grains.

## Reported nutrient content of meat-containing ITFs (Aim 2)

**Table 3** displays energy, protein, iron, and zinc present in infant, toddler, and young child products and meat snacks per serving and per 100 g. **Table 4** displays the same nutrients within infant, toddler, and young child foods by meat type.

**Table 3. Energy, protein, iron, and zinc per serving and per 100 grams present in infant and toddler products.**

| | Energy (kcal) | | | | | Protein (g) | | | | | Iron (mg)[1] | | | | | Zinc (mg)[1,2] | | | | |
|---|---|---|---|---|---|---|---|---|---|---|---|---|---|---|---|---|---|---|---|---|
| | | Per serving | | Per 100 g | | | Per serving | | Per 100 g | | | Per serving | | Per 100 g | | | Per serving | | Per 100 g | |
| | *n* | Mdn | IQR | Mdn | IQR | *n* | Mdn | IQR | Mdn | IQR | *n* | Mdn | IQR | Mdn | IQR | *n* | Mdn | IQR | Mdn | IQR |
| Infant (RACC 110g) | 65 | 80.0 | 20.0 | 78.1 | 29.0 | 65 | 4.0 | 3.0 | 3.5 | 3.6 | 51 | 0.5 | 0.8 | 0.5 | 0.7 | 14 | 0.4 | 0.5 | 0.4 | 0.7 |
| Toddler (RACC 170g) | 38 | 115.0 | 72.5 | 88.0 | 36.1 | 38 | 6.0 | 5.5 | 6.0 | 5.7 | 35 | 1.0 | 0.6 | 0.7 | 0.7 | 1 | 0.3 | – | 0.2 | – |
| Young Child (No RACC) | 55 | 250.0 | 80.0 | 149.3 | 55.4 | 55 | 14.0 | 6.0 | 8.9 | 3.1 | 55 | 1.0 | 1.0 | 0.9 | 0.8 | 0 | – | – | – | – |
| Meat Snacks (RACC 55g) | 7 | 130.0 | 40.0 | 238.1 | 109.2 | 7 | 8.0 | 3.0 | 12.7 | 3.6 | 7 | 0.6 | 0.5 | 1.1 | 0.4 | 2 | 1.2 | – | 1.6 | – |

*Note*. Mdn = Median; IQR = Interquartile range; RACC = reference amount customarily consumed per eating occasion; IQRs cannot be calculated when a product category contains 3 products or fewer

[1]The estimated average requirement (EAR) for iron is 6.9 mg/day for infants and 3.0 mg/day for toddlers. The EAR for zinc is 2.5 mg/day for both infants and toddlers (see reference 3).

[2]Information on zinc content was not available for most products.

**Table 4. Energy, protein, iron, and zinc per serving and per 100 grams present in infant and toddler products by meat type.**

| | Energy (kcal) | | | | | Protein (g) | | | | | Iron (mg)[1] | | | | | Zinc (mg)[1,2] | | | | |
|---|---|---|---|---|---|---|---|---|---|---|---|---|---|---|---|---|---|---|---|---|
| | | Per serving | | Per 100 g | | | Per serving | | Per 100 g | | | Per serving | | Per 100 g | | | Per serving | | Per 100 g | |
| | n | Mdn | IQR | Mdn | IQR | n | Mdn | IQR | Mdn | IQR | n | Mdn | IQR | Mdn | IQR | n | Mdn | IQR | Mdn | IQR |
| Infant[3] | 65 | | | | | | | | | | | | | | | | | | | |
| Beef | 16 | 85.0 | 30.0 | 78.9 | 32.2 | 16 | 5.0 | 4.8 | 4.5 | 6.8 | 13 | 0.8 | 0.7 | 0.8 | 0.6 | 3 | 0.9 | – | 0.8 | – |
| Chicken | 28 | 80.0 | 20.0 | 70.8 | 26.5 | 28 | 4.0 | 3.8 | 3.1 | 3.5 | 20 | 0.4 | 0.5 | 0.4 | 0.6 | 6 | 0.3 | 0.3 | 0.3 | 0.3 |
| Pork/ham | 2 | 65.0 | – | 100.2 | – | 2 | 6.5 | – | 10.0 | – | 2 | 0.3 | – | 0.5 | – | 1 | 0.8 | – | 1.1 | – |
| Seafood | 4 | 105.0 | 17.5 | 87.2 | 6.6 | 4 | 4.5 | 1.0 | 3.7 | 1.6 | 4 | 0.6 | 0.8 | 0.5 | 0.9 | 0 | – | – | – | – |
| Turkey | 15 | 80.0 | 20.0 | 70.8 | 39.1 | 15 | 3.0 | 4.0 | 2.7 | 4.0 | 12 | 0.6 | 0.6 | 0.5 | 0.5 | 4 | 0.4 | 0.8 | 0.4 | 1.2 |
| Toddler[3] | 38 | | | | | | | | | | | | | | | | | | | |
| Beef | 9 | 130.0 | 50.0 | 113.5 | 50.2 | 9 | 7.0 | 4.5 | 5.7 | 6.6 | 9 | 1.0 | 0.9 | 0.9 | 0.9 | 0 | – | – | – | – |
| Chicken | 18 | 115.0 | 65.0 | 85.3 | 36.7 | 18 | 6.0 | 7.5 | 6.7 | 7.0 | 17 | 0.8 | 0.8 | 0.7 | 0.6 | 1 | 0.3 | – | 0.2 | – |
| Pork/ham | 2 | 105.0 | – | 99.8 | – | 2 | 9.0 | – | 8.9 | – | 2 | 1.1 | – | 0.9 | – | 0 | – | – | – | – |
| Seafood | 2 | 165.0 | – | 101.8 | – | 2 | 9.5 | – | 5.7 | – | 1 | 2.0 | – | 1.1 | – | 0 | – | – | – | – |
| Turkey | 7 | 110.0 | 40.0 | 85.9 | 18.4 | 7 | 5.0 | 4.0 | 3.9 | 3.8 | 6 | 1.0 | 1.4 | 0.9 | 0.8 | 0 | – | – | – | – |
| Young Child[4] | 55 | | | | | | | | | | | | | | | | | | | |
| Beef | 9 | 260.0 | 115.0 | 146.9 | 57.9 | 9 | 14.0 | 7.5 | 7.6 | 2.5 | 9 | 2.0 | 2.0 | 1.0 | 1.0 | 0 | – | – | – | – |
| Chicken | 29 | 270.0 | 105.0 | 146.7 | 61.1 | 29 | 15.0 | 7.0 | 9.1 | 3.1 | 29 | 1.0 | 1.0 | 0.9 | 0.9 | 0 | – | – | – | – |
| Pork/ham | 7 | 360.0 | 140.0 | 171.9 | 26.0 | 7 | 18.0 | 7.0 | 10.6 | 4.0 | 7 | 2.0 | 1.0 | 1.5 | 0.7 | 0 | – | – | – | – |
| Seafood | 1 | 240.0 | – | 124.4 | – | 1 | 19.0 | – | 9.8 | – | 1 | 1.0 | – | 0.5 | – | 0 | – | – | – | – |
| Turkey | 11 | 240.0 | 70.0 | 156.3 | 74.1 | 11 | 12.0 | 6.0 | 8.9 | 2.9 | 11 | 1.0 | 0.6 | 0.9 | 0.5 | 0 | – | – | – | – |

Mdn = Median; IQR = Interquartile range

[1]The estimated average requirement (EAR) for iron is 6.9 mg/day for infants and 3.0 mg/day for toddlers. The EAR for zinc is 2.5 mg/day for both infants and toddlers (see reference 3).

[2]Information for zinc content was not available for most products.

[3]The reference amount customarily consumed per eating occasion (RACC) is 110 g for infant products and 170 g for toddler products (see reference 10). [4]Products containing more than one meat (*n* = 2) are represented in more than one column.

## Energy

Infant and toddler products contained similar amounts of energy per 100 g and both contained less energy than young child products. Meat snacks contained the most energy per 100 g, but it is important to note that 100 g is almost twice the RACC for meat snacks (55 g), so this amount is unlikely to be consumed in one eating occasion. When considering the amount of energy per serving, meat snacks were comparable to toddler products.

## Protein

Meat snacks contained the most protein per 100 g, followed by young child and toddler products; however, meat snacks contained a similar amount of protein per serving compared to toddler products. Infant products contained less protein per 100 g and per serving than all other product categories.

## Iron

Both toddler and young child products contained greater amounts of iron per 100 g compared to infant products. Meat snacks contained the most iron (1.1 ± 0.4 mg) per 100 g compared to other products; however, meat snacks contained similar amounts of iron per serving (0.6 ± 0.5 mg) compared to infant products (0.5 ± 0.8 mg).

Within infant products ($n$ = 65), more iron was found in products containing beef. Similar amounts of iron (per 100 g) were found among products containing chicken, pork/ham, seafood, and turkey. Within toddler products ($n$ = 38), more iron was found in products containing seafood, followed by beef, turkey, and pork. Slightly less iron was found in products containing chicken. Finally, within young child products ($n$ = 55), more iron was found in products containing pork/ham, followed by beef, chicken, and turkey. Less iron was found in products containing seafood.

According to FDA definitions [12, 13], none of the infant products were categorized as a "good" source of iron. Of the 38 toddler products, 20 (53%) were considered to be a "good" source of iron since they included 10–19% of the toddler DV for iron. When using the EAR, 19 (29%) infant products and 29 (76%) toddler products contained 10–19% of the EAR for iron for infants and toddlers, respectively.

### Zinc

Zinc was infrequently reported on product labels ($n$ = 14 infant products; $n$ = 1 toddler product; $n$ = 0 young child products; and $n$ = 2 meat snacks). Within infant products reporting zinc content, more zinc was found in the product containing pork, followed by products containing beef. Less zinc was found in products containing chicken and turkey. No comparisons are included for toddler or young child products given that so few products reported the amount of zinc contained in the product.

## Discussion

Given the critical importance of iron and zinc during infancy and toddlerhood, it is necessary to understand whether and how infants and toddlers can meet recommended intakes of these nutrients. In the present study, we examined the prevalence, variety, and nutrient contents of meat-containing ITFs sold in the U.S. to determine which of these products (e.g., by meat type, target audience) are most promising in helping infants and toddlers meet their nutrient needs. We found that many products in our database contained low amounts of iron and zinc per serving and per 100 g. Furthermore, infant products tended to contain lower amounts of iron than toddler and young child products and no infant products were a "good" source of iron based on the DV for infants. Of note, more products met the definition of a "good source" based upon the EAR for infants and toddlers (the DV for iron is higher than the EAR as it is calculated based upon assumptions of a lower bioavailability of iron from foods). Regardless, the limited number of products meeting the definition for "good source" could be problematic given that infants exclusively fed human milk should consume iron at greater levels than toddlers and young children.

Very few products in our database reported zinc on their Nutrition Facts panels, which makes it challenging for caregivers to know whether products are a good source of this important nutrient. In this study, so few products reported zinc on product labels that we could not make comparisons among products according to zinc content. Lastly, we report the amount of energy and protein per serving and per 100 g for infant, toddler, and young child products because parents often report concerns about infant growth, particularly growth at the lowest percentiles on growth charts [14]. However, despite parental concerns, total energy and protein intakes are not of concern for most infants and toddlers based on national estimates [15]. Therefore, we focus our discussion primarily on iron.

Based on the results of the present study, it is unclear how commercial ITFs containing meat can be ingested at levels required to meet the recommended intakes for both iron and zinc. However, concerns about adequate iron and zinc intakes are not unique to infants/

toddlers consuming commercially available ITFs. Abrams et al. [16] calculated iron absorption for infants during complementary feeding (6–12 months of age) who were receiving human milk, formula, or both, in addition to solid foods. Notably, the sample was drawn from the FITS which has been reported to include few infants consuming baby food meats (4% of infants; see reference 2). Calculated iron absorption in this sample was inadequate for over half of infants (ranging from 20% of formula-fed infants to almost 96% of infants fed human milk). Heme iron contributed very little to iron absorption in all groups, with chicken and turkey being the most consumed meats. Instead, the greatest contributor to iron consumption for all groups was non-heme iron from fortified foods: specifically grains (i.e., infant cereal) for all groups and formula for groups that received it [16]. Without accompanying clinical indicators of iron status, it is unclear whether a diet of primarily commercial ITFs, particularly including products with heme iron, can impact the adequacy of iron absorption and the prevalence of iron deficiency and anemia.

Caregivers feeding infants and toddlers could benefit from knowing which types of commercial ITFs tend to contain greater amounts of iron (and zinc) compared to other products. Several factors contributed to the overall iron content of the product in the present analysis, including the type of meat and other ingredients included. In this analysis, the types of meats contained in products with greater amounts of iron differed by the target age group of the products. Infant products containing beef, toddler products containing seafood, and young child products containing pork/ham all contained greater amounts of iron compared to products with other types of meat targeted toward the same age groups. Conversely, products containing chicken within all three age categories tended to contain lower amounts of iron compared to products containing other meat types. Chicken was also the most included meat among all ITFs and was present in almost half of the available meat-containing products for infants, toddlers, and young children. When these results are considered together, it seems possible that the high prevalence of products containing chicken, a meat that has lower amounts of iron compared to other meats [17], may contribute, in part, to the low levels of iron found in ITFs. However, it is important to consider the other ingredients included in products, which also contribute to each product's total iron content.

Very few ITFs available in the U.S. contained meat as the sole ingredient in the product. As a result, the iron content of most available meat-containing ITFs is influenced by both the type of meat it includes, as well as the presence of other ingredients (e.g., grains, vegetables). The most common ingredients included alongside meat across all age categories in our database were red/orange vegetables (e.g., butternut squash, carrots), other vegetables (e.g., broccoli, peas), and grains (e.g., pasta). Some of these ingredients are iron-rich (i.e., dark green vegetables), some ingredients (i.e., pasta) include iron due to fortification, and some ingredients (i.e., carrots) contain relatively less iron. Each of these additional ingredients contains non-heme iron, which is less bioavailable than the heme iron found in meat [3]. Thus, while these additional ingredients contribute to the overall iron content of the product, the iron they provide will not be as readily absorbed by infants/toddlers compared to the iron provided by meat. Additional ingredients (i.e., foods high in Vitamin C) can also impact the bioavailability of iron in a product, though few products contained these ingredients (e.g., citrus fruits, bell peppers, cruciferous vegetables). In sum, the presence of specific meat types and other ingredients both contribute to the overall iron content and its bioavailability in the product. However, this information is not available to consumers because ingredient lists only report the relative contribution (by weight) of each ingredient in the product and Nutrition Facts panels do not include information on bioavailability of iron or other nutrients in the product.

Caregivers feeding infants and toddlers who are concerned about iron and zinc intakes for their child are faced with a difficult dilemma about how to meet these needs. For families who

are willing to serve meat to their child, a wider array of commercial ITFs rich in heme iron could be useful. Manufacturers should be encouraged to develop a wider selection of commercial products with meats rich in heme iron (e.g., beef) or increase the relative contribution of meat, and therefore heme iron, in available products. One important consideration is the ability to appeal to both caregivers and infants and to make products which are sufficiently palatable to both [18]. Prior research indicates that infants' tolerance and acceptance of puréed beef and infant cereal are comparable according to parent perceptions [19]. Future research should explore whether these findings can be replicated for infant and toddler products containing other meat types and investigate how parent acceptance of these products impacts their purchasing behaviors.

Other commercial ITFs that do not contain meat could also be utilized to help infants and toddlers meet recommended intakes for iron and zinc. Iron-fortified infant cereals tend to be excellent sources of iron [6]. Other products, such as cereal bars and breakfast pastries, are also "excellent" sources of iron for infants and toddlers, though caregivers should be cautioned that these products may also contain high levels of sodium and/or sugars [9]. Another strategy is to offer homemade foods. Caregivers who are willing and able to prepare homemade foods for their infants and toddlers should be encouraged to offer iron- and zinc-rich foods such as red meat, poultry, tofu, beans/lentils, and some vegetables. However, offering homemade foods may not be feasible for some caregivers due to a variety of barriers including: 1) knowledge about which foods contain high levels of available iron and zinc (which requires nutrition literacy; [20]), 2) ability to purchase these foods (which involves availability of these foods at grocery stores or markets, affordability of these products, and eligibility for coverage by WIC; [21–23]), and 3) ability and willingness to prepare these foods (which is affected by caregiver and family food likes and dislikes, food literacy and cooking skills, and having the necessary time and resources; [24, 25]). In sum, there is a great need for solutions that provide adequate amounts of iron and zinc for infants and toddlers, but that are also affordable, palatable, culturally appropriate, and convenient for caregivers to purchase or prepare, particularly in light of the recommendations emphasizing the need to move towards more plant-based diets [26].

The analysis presented here is based on a comprehensive search of commercial meat-containing ITFs available in the United States. However, it is possible that relevant companies or products were not included in our database due to lack of an online presence. It is also possible that company websites were outdated, so they may not have reflected current product offerings. Furthermore, although our database includes the total iron and zinc content reported on Nutrition Facts panels, these values do not reflect the bioavailability of these nutrients in products, which is affected by factors such as type of iron (heme vs. non-heme) and the presence of other ingredients. Relatedly, although we focused on the nutrient content of products containing meat, other products and ingredients, such as grains, vegetables (including beans), and infant and toddler milks, also contribute to overall iron intakes. As a result, we are unable to determine the relative contribution of meat vs. other ingredients to children's daily intakes of iron and zinc. Another limitation to our analysis is that few products reported the amount of zinc contained in the product on Nutrition Facts panels. Currently, manufacturers are not required to list zinc on product labels unless it has been added to the food [27]. Because so few products reported zinc content, we were unable to thoroughly analyze the presence of zinc in meat-containing products. Further, the products that did report zinc may not be representative of the typical amount of zinc found in similar types of products. Although the lack of zinc reporting is a limitation to our analysis, it is a larger concern for caregivers who, based on product labels alone, are not able to tell how much zinc a product may provide. To address this issue, manufacturers and regulatory agencies could be strongly encouraged or required to include zinc content on Nutrition Facts labels. Finally, our database and analysis reflect the

availability of products containing meat, not the consumption of these products by infants and toddlers.

## Conclusions

Based on the results of this analysis, meeting the current iron and zinc recommendations from solid foods during the complementary feeding period, particularly for infants fed human milk, may be challenging if commercial ITFs containing meat are the primary source of these nutrients. Caregivers feeding these products to their infants and toddlers should be encouraged to examine Nutrition Facts panels and select products that report zinc content, as well as products that contain greater amounts of both iron and zinc. Even with this approach, it is unclear whether these products are sufficient to meet the estimated iron and zinc needs of infants and toddlers. A potential strategy for improvement is to encourage manufacturers to expand the availability of iron- and zinc-rich ITFs that are commercially available, particularly those that contain greater amounts of heme iron which is most bioavailable. Additionally, caregivers who are willing and able to offer their child homemade iron- and zinc-rich foods should be offered resources that support and encourage them to do so if it fits their family needs and lifestyle. Finally, because inadequate iron and zinc intake can affect all infants and toddlers, regardless of their primary sources of these nutrients, the American Academy of Pediatrics recommends universal screening for anemia at 12 months of age, including hemoglobin concentration and risk factors associated with iron deficiency and iron deficiency anemia [28]. However, increased monitoring of iron status (iron deficiency without anemia) would be of benefit in determining whether iron needs are being met for young children [4]. Because there are no universal screening methods for detecting inadequate zinc status [29, 30], better information for caregivers to monitor their child's zinc intake is needed.

## Author Contributions

**Conceptualization:** Kameron J. Moding, Catherine A. Forestell, Susan L. Johnson.

**Data curation:** Kameron J. Moding, Megan C. Lawless, Susan L. Johnson.

**Formal analysis:** Kameron J. Moding.

**Funding acquisition:** Catherine A. Forestell, Susan L. Johnson.

**Investigation:** Kameron J. Moding, Catherine A. Forestell, Katherine J. Barrett, Susan L. Johnson.

**Methodology:** Kameron J. Moding, Megan C. Lawless, Catherine A. Forestell, Katherine J. Barrett, Susan L. Johnson.

**Project administration:** Kameron J. Moding, Katherine J. Barrett, Susan L. Johnson.

**Resources:** Susan L. Johnson.

**Supervision:** Kameron J. Moding, Susan L. Johnson.

**Writing – original draft:** Kameron J. Moding.

**Writing – review & editing:** Megan C. Lawless, Catherine A. Forestell, Katherine J. Barrett, Susan L. Johnson.

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
