## [Decision Letter · Decision Letter 0]

21 May 2024

PONE-D-23-41996Prevalence, Variety, and Iron and Zinc Content of Commercial Infant and Toddler Foods Sold in the United States that Contain MeatPLOS ONE

Dear Dr. Johnson,

Thank you for submitting your manuscript to PLOS ONE. After careful consideration, we feel that it has merit but does not fully meet PLOS ONE’s publication criteria as it currently stands. Therefore, we invite you to submit a revised version of the manuscript that addresses the points raised during the review process.

**Your Manuscript needs MINOR revision, please resubmit after addressing the reviewer's comments. **

We look forward to receiving your revised manuscript.

Kind regards,

Sajid Bashir Soofi

Academic Editor

PLOS ONE

Journal Requirements:

 [Susan L. Johnson, Kameron J. Moding, Catherine Forestell, and Katherine J. Barrett report that financial support was provided by National Cattlemen’s Beef Association with funding from the Checkoff Program for this study.  Contract # 25B0508.].  

Please respond by return e-mail so that we can amend your financial disclosure and competing interests on your behalf.

[Susan L. Johnson, Kameron J. Moding, Catherine Forestell, and Katherine J. Barrett report that financial support was provided by National Cattlemen’s Beef Association with funding from the Checkoff Program for this study.  Contract # 25B0508.]. 

Please respond by return email with your amended Competing Interests Statement and we will change the online submission form on your behalf.

5. In the online submission form, you indicated that [The data underlying the results presented in the study are available from Kameron J Moding KJModing@purdue.edu]. 

Additional Editor Comments:

thanks for submitting your manuscript to Plos One. Your paper needs MINOR revision, Please address the comments of the Reviewer point by point and resubmit your manuscript within 15 days.

Reviewers' comments:

Reviewer's Responses to Questions

**Comments to the Author**

1. Is the manuscript technically sound, and do the data support the conclusions?

Reviewer #1: Yes

Reviewer #2: Yes

2. Has the statistical analysis been performed appropriately and rigorously? 

Reviewer #1: Yes

Reviewer #2: Yes

3. Have the authors made all data underlying the findings in their manuscript fully available?

Reviewer #1: Yes

Reviewer #2: Yes

4. Is the manuscript presented in an intelligible fashion and written in standard English?

Reviewer #1: Yes

Reviewer #2: Yes

5. Review Comments to the Author

Reviewer #1: The study provides valuable insights into the nutrient composition of commercially available infant and toddler foods (ITFs) containing meat, focusing on iron and zinc content in the US. The categorization of products by intended age/stage and meat type offers a structured analysis of nutrient variability in these food items. Findings highlight the potential challenges in meeting iron and zinc requirements in infants during the complementary feeding period, when relying on commercial infant and toddler foods containing meat, emphasizing the importance of nutrient-rich food choices for infants and toddlers.

This study has discussed an important aspect of child health and nutrition by assessing the nutrient content of ITFs. However, the small sample size and limited variety of meat products may affect the generalizability of the results. Nonetheless, it provides a valuable direction for future research and further exploration. Future studies should aim to increase the sample size and incorporate a broader range of meat products to enhance the robustness and generalizability of the findings.

Some minor comments from my end:

- Line 27 – 31: Does it imply that human milk alone may not always provide sufficient iron and zinc for infants, and therefore, complementary foods with these nutrients are important, especially during the transition to solid foods? Please decipher.

- Line 157 – 168: I would encourage the authors to include a table for the description data.

- I suggest that the authors include a discussion on the implications of the study findings for public health policies, particularly advocating for improved regulations or guidelines to enhance the nutritional quality of infant and toddler foods.

Reviewer #2: Well, written manuscript on an important public health issue especially in context to micronutrient requirements in growing children.

The methodology is sound and scientific.

In the current scenario with many mothers using ready-made food, this study provides rationale of supplementation of zinc and iron in addition to complimentary diet

6. PLOS authors have the option to publish the peer review history of their article (what does this mean?). If published, this will include your full peer review and any attached files.

Reviewer #1: **Yes: **Ahmad Khan

Reviewer #2: **Yes: **Shabina Ariff

---

## [Author Response · Author response to Decision Letter 0]

11 Jun 2024

Reviewer #1: The study provides valuable insights into the nutrient composition of commercially available infant and toddler foods (ITFs) containing meat, focusing on iron and zinc content in the US. The categorization of products by intended age/stage and meat type offers a structured analysis of nutrient variability in these food items. Findings highlight the potential challenges in meeting iron and zinc requirements in infants during the complementary feeding period, when relying on commercial infant and toddler foods containing meat, emphasizing the importance of nutrient-rich food choices for infants and toddlers.

This study has discussed an important aspect of child health and nutrition by assessing the nutrient content of ITFs. However, the small sample size and limited variety of meat products may affect the generalizability of the results. Nonetheless, it provides a valuable direction for future research and further exploration. Future studies should aim to increase the sample size and incorporate a broader range of meat products to enhance the robustness and generalizability of the findings. Thank you for the positive comments about our manuscript. 

We agree that the limited number of meat-containing products may affect the generalizability of the results. However, the limited number and variety of products in our analysis reflects the limited number of infant and toddler food products containing meat that are currently available for purchase in the United States. Increasing the number of products available for analysis would require companies to expand their offerings of foods available for infants and toddlers that contain meat. 

Some minor comments from my end:

- Line 27 – 31: Does it imply that human milk alone may not always provide sufficient iron and zinc for infants, and therefore, complementary foods with these nutrients are important, especially during the transition to solid foods? Please decipher.

Yes, your interpretation is correct. Human milk alone does not provide sufficient iron and zinc for infants by 6 months of age, so complementary foods with these nutrients are needed. The language in the manuscript has been revised to reflect this point (Lines 25- 28). 

- Line 157 – 168: I would encourage the authors to include a table for the description data. We now include these data in a table (Table 2). Since all data are now included in this table, we have removed some sentences to streamline this paragraph and to focus on the other ingredient types that were most commonly included in each product category. All statistics have also been removed from the text of this paragraph since they are included in the new table.

- I suggest that the authors include a discussion on the implications of the study findings for public health policies, particularly advocating for improved regulations or guidelines to enhance the nutritional quality of infant and toddler foods. We appreciate this suggestion. We now directly state two recommendations: 1) Manufacturers should be encouraged to develop a wider selection of commercial products with heme iron and/or increase the relative contribution of meat in products (Lines 306-308), and 2) Manufacturers and regulatory agencies could be strongly encouraged or required to include zinc on Nutrition Facts panels (Lines 351-353). 

In responding to this comment in the manuscript, two new references were added to the text. The two new references are now included in the reference list.

Reviewer #2: Well, written manuscript on an important public health issue especially in context to micronutrient requirements in growing children.

The methodology is sound and scientific.

In the current scenario with many mothers using ready-made food, this study provides rationale of supplementation of zinc and iron in addition to complimentary diet. Thank you for the positive comments about our manuscript.

---

## [Editor Report · Decision Letter 1]

13 Jun 2024

Prevalence, Variety, and Iron and Zinc Content of Commercial Infant and Toddler Foods Sold in the United States that Contain Meat

PONE-D-23-41996R1

Hi Dr. Johnson,

We’re pleased to inform you that your manuscript has been judged scientifically suitable for publication and will be formally accepted for publication once it meets all outstanding technical requirements.

Kind regards,

Sajid Bashir Soofi

Academic Editor

PLOS ONE

Additional Editor Comments (optional):

Dear Authors, Thank you for addressing the comments. Congratulations now your paper is accepted for publication.
---

## [Editor Report · Acceptance letter]

20 Jun 2024

PONE-D-23-41996R1 

PLOS ONE

Dear Dr. Johnson, 

I'm pleased to inform you that your manuscript has been deemed suitable for publication in PLOS ONE. Congratulations! Your manuscript is now being handed over to our production team.

Kind regards, 

on behalf of

Professor Sajid Bashir Soofi 

Academic Editor

PLOS ONE